# Placing Management of Sunflower Downy Mildew (*Plasmopara halstedii* (Farl.) Berl. et de Toni) under an Integrated Pest Management (IPM) System Approach: Challenges and New Perspectives

Rita Bán *, József Kiss, Zoltán Pálinkás and Katalin Körösi

Department of Integrated Plant Protection, Institute of Plant Protection, Hungarian University of Agriculture and Life Sciences, H-2103 Godollo, Hungary; jozsef.kiss@uni-mate.hu (J.K.); palinkas.zoltan@uni-mate.hu (Z.P.); korosi.katalin.orsolya@uni-mate.hu (K.K.)
* Correspondence: ban.rita@uni-mate.hu

**Abstract:** Sunflower is one of the major oil crops in the world. Diseases such as sunflower downy mildew (*Plasmopara halstedii* (Farl.) Berl. et de Toni) constitute a significant risk factor during sunflower production. Integrated pest management (IPM) is considered an essential tool against sunflower downy mildew; however, the pathogen variability repeatedly affects the efficacy of control measures. This article evaluates some vital elements of the management of sunflower downy mildew disease and analyzes current challenges. In addition, we outlined the options for the future integration of recent research and achievements related to sunflower downy mildew to achieve more sustainable sunflower production. Finally, a SWOT analysis was performed to consider internal factors, such as strengths (S) and weaknesses (W), and external factors, such as opportunities (O) and threats (T) connected to the topic.

**Keywords:** *Plasmopara halstedii*; integrated pest management; sunflower downy mildew; SWOT analysis; sunflower; innovative techniques; genetic resistance; crop rotation; seed treatment; pathotype



## 1. Current Situation and Plant Health Aspects of Sunflower Cultivation

Sunflower is among the world's major oil crops, along with soybean, palm, and oilseed rape [1]. At present, we are experiencing a significant increase in interest in sunflower cultivation due to increased global and regional market demands and farming profitability. First, the high revenue from this crop is a crucial driving force for farmers and companies, as indicated by the forecasts, which show an annual growth rate of 7.68% in income from edible oil production by 2027 [2]. Another reason for increasing sunflower cultivation is the global impact of the situation in Ukraine and Russia. These countries account for more than half of the world's sunflower production [3]. The beneficial properties of this plant also support the rising interest in sunflower cultivation, and thus its adaptability to climatic changes and different agricultural systems [4–6].

As plant diseases constitute a significant risk in sunflower cultivation, the intensification of cultivation or increase in acreages may pose severe challenges for farmers, plant breeders, and crop protection experts. Furthermore, plant pathogens, such as *Plasmopara halstedii*, the causal agent of sunflower downy mildew, play a crucial role in the yield decline in sunflower cultivation worldwide [7–9]. Although environmental factors are assessed to be less prone to diseases in general, the continuous change in pathogenic dominance and variability is likely to create a new situation in integrated pest management (IPM) in sunflower production [10,11]. Therefore, this article evaluates selected vital elements of the management of sunflower downy mildew disease and analyzes current challenges. In addition, we outline the options for the future integration of recent research results and achievements related to sunflower downy mildew to achieve a more sustainable sunflower production.

## 2. Downy Mildew as a Major Threat to Sunflowers

### 2.1. Significance of Sunflower Downy Mildew

Sunflower downy mildew is a long-known global disease [12]. It is caused by *Plasmopara halstedii* (Farl.) Berl. et de Toni, an oomycete that is widespread worldwide. According to Gulya et al. [13], the damage caused by this pathogen may reach 100% when the disease occurs in patches rather than sporadically in the field. To date, *P. halstedii* has been classified as a quarantine pest, but due to its significant prevalence, the pathogen belongs to the regulated non-quarantine pests (RNQPs), as of 2019, in most European countries [14–16].

IPM is fundamental to managing sunflower downy mildew. However, the pathogen variability repeatedly affects the efficacy of control measures [17–19]. This means that there are currently about 50 variants of the pathogen, known as pathotypes (races), in the world, and this number is growing significantly every year [9,20]. Furthermore, the dominance of the different pathotypes is constantly changing, with new and more aggressive variants emerging [13,21]. Notably, less aggressive pathotypes identified earlier continue to predominate in the *P. halstedii* population, presumably due to their high pathogenic fitness, making the situation significantly more complex [9].

One of the reasons for the evolution of several pathotypes of *P. halstedii* is the continuous development of dominant resistance genes incorporated into new sunflower hybrids, which exerts an intense selection pressure on the pathogen population [18,22]. Another reason for the variability is the hybridization or the prevailing asexual reproduction within *P. halstedii* metapopulations [23]. This phenomenon, along with multiple local introductions of variants via seed transfer, considerably accelerates the emergence of new pathotypes of sunflower downy mildew [24].

### 2.2. Biological Aspects of IPM against Plasmopara halstedii: Symptoms, Signs, and Life Cycle

In addition to the variability, the symptoms caused by this pathogen also contribute to the threat of the disease. *Plasmopara halstedii* most often infects the root of young plants with zoospores that reach the underground plant organs via chemotaxis [25]. Once attached to the roots, a zoospore loses its flagella and encysts by secreting a wall. The resulting cystospore penetrates the plant tissues with a germ tube [26]. The mycelium of the pathogen spreads through the intercellular spaces to the above-ground parts of the plant, releasing modified hyphae (haustoria) into the cells for nutrition. The resulting symptoms are the dwarfing of diseased plants (Figure 1a), leaf chlorosis along the veins (Figure 1b), erected heads, and sterile or non-viable seeds. Heavy infestation at a very early sunflower growth stage can also lead to plant damping-off [27].

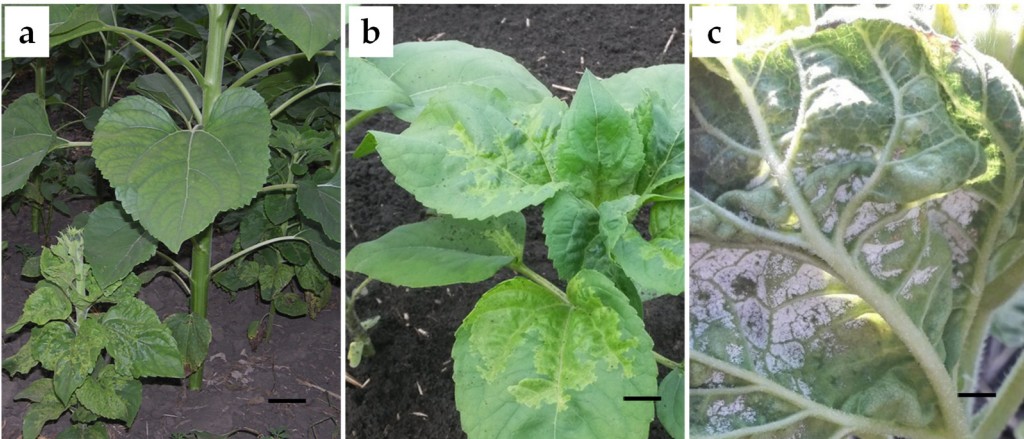

**Figure 1.** Symptoms and signs of sunflower downy mildew. (**a**) Dwarfing of the diseased plants (Photo: R. Bán); (**b**) leaf chlorosis along the veins (Photo: Z. Pálinkás); and (**c**) white coating (sporangiophores and sporangia) on the back of the leaf (Photo: K. Körösi). Scale bars represent 10 cm (**a**), 2.4 cm (**b**), and 1.25 cm (**c**).

The characteristic signs of the pathogen are a white coating on the abaxial surface of leaves (Figure 1c), which consists of the asexual reproductive organs as sporangiophores and sporangia with the zoospores. Secondary infection by sporangia (zoospores) can also occur within the vegetation period, which causes primarily local symptoms, such as angular leaf lesions. While this phenomenon is less frequent than primary infection, even the secondary infection can cause a significant yield loss [28]. However, local symptoms can turn to systemic infection in upper plant parts, resulting in latent seed contamination by the pathogen [29]. Contaminated seeds can disperse *P. halstedii* for considerable distances.

Due to the biotrophic nature of the pathogen, many diseased plants survive until harvest, serving as a source for the disease to spread. As a result of sexual reproduction (oogamy), oospores develop in the stalk towards the end of the growing season. The oospores are the resting organs of *P. halstedii* that can remain viable in the soil for up to 10 years [22,30]. *Plasmopara halstedii* is primarily a soil-borne pathogen but can survive in seeds (seed-borne), host weeds, and sunflower crop residues, such as oospores and mycelium.

### 2.3. Predisposing Environmental Conditions for Disease Development

In addition to a virulent pathogen and a susceptible host plant, optimal environmental conditions are essential for successful infection. Cool (10 to 15 °C) and rainy weather around the sowing of sunflowers are favorable for downy mildew. Air temperatures above this level inhibit the spread of the pathogen mycelium inside the plant [13,31]. Additionally, higher soil temperature (above 20 °C) following sunflower sowing significantly inhibits the germination of oospores and infection even in high rainfall and irrigation [13,32]. According to Deabeke et al. [7], 50 mm of precipitation within ten days around sowing time is essential for this disease establishment with a soil temperature of 10 °C at least.

The environment in the crop stand created by cultural practices has a crucial role in disease development. Denser and inadequately fertilized (depending on regional recommendations), sunflower stands are more susceptible to this disease. Furthermore, timely narrow crop rotation (re-planting within four years) and insufficient host plant resistance can also contribute to the disease's spread. Leaving infested plant residues and volunteers in the crop field predisposes the disease, as the pathogen can survive on them [8,9]. Several weed species, including those of genera *Ambrosia*, *Iva*, and *Xanthium*, are host plants of *P. halstedii*, which allow the pathogen to persist and increase the relative humidity of the plant stand, facilitating infection [33].

## 3. Integrated Pest Management against Sunflower Downy Mildew

### 3.1. IPM as a Holistic Approach

Integrated pest management was defined by the European Commission Framework Directive on the Sustainable Use of Pesticides [34] as follows: "Integrated pest management means careful consideration of all available plant protection methods and subsequent integration of appropriate measures that discourage the development of populations of harmful organisms and keep the use of plant protection products and other forms of intervention to levels that are economically and ecologically justified and reduce or minimize risks to human health and the environment. Integrated pest management emphasizes the growth of a healthy crop with the least possible disruption to agro-ecosystems and encourages natural pest control mechanisms".

IPM is a holistic approach relying on the multiplicity of and synergy among various plant protection methods related to different cropping systems rather than individual plants and single pests [35]. IPM also considers site-specific factors, including regional cropping patterns, the surrounding landscape, semi-natural habitats, and pest pressure. Furthermore, implementing non-technical factors, such as market, economic factors, and training facilities, are among the main goals of IPM. The eight principles of IPM consist of:

1. Prevention and suppression;
2. Monitoring;

3. Decision-making process;
4–7. Control options;
8. Evaluation.

Prevention implies developing production systems that minimize the economic losses caused by pests, while suppression is to decrease their incidence and severity. Principle 1 implies the integration of control mechanisms, such as crop rotation, using resistant cultivars, increasing the diversity within and around the field, and other tactics, into management strategies. Principle 2 covers monitoring based on regular pest surveillance by the farmer and using forecasting systems. For decision-making process (Principle 3), considering previous tactics and the possible application of thresholds to assess pest pressure is essential. When intervention (Principles 4–7) is necessary, non-chemical methods, such as the application of biological control, are preferred. Moreover, carefully considering reduced pesticide usage and anti-resistant strategies must be conducted before using chemicals. Finally, the evaluation of crop protection measures (Principle 8) with a multi-season approach (covering all crops in the rotation) is indispensable for integrated plant protection [7,35–37]. Figure 2 summarizes the methods used in the integrated disease management of sunflower downy mildew discussed in the following sections.

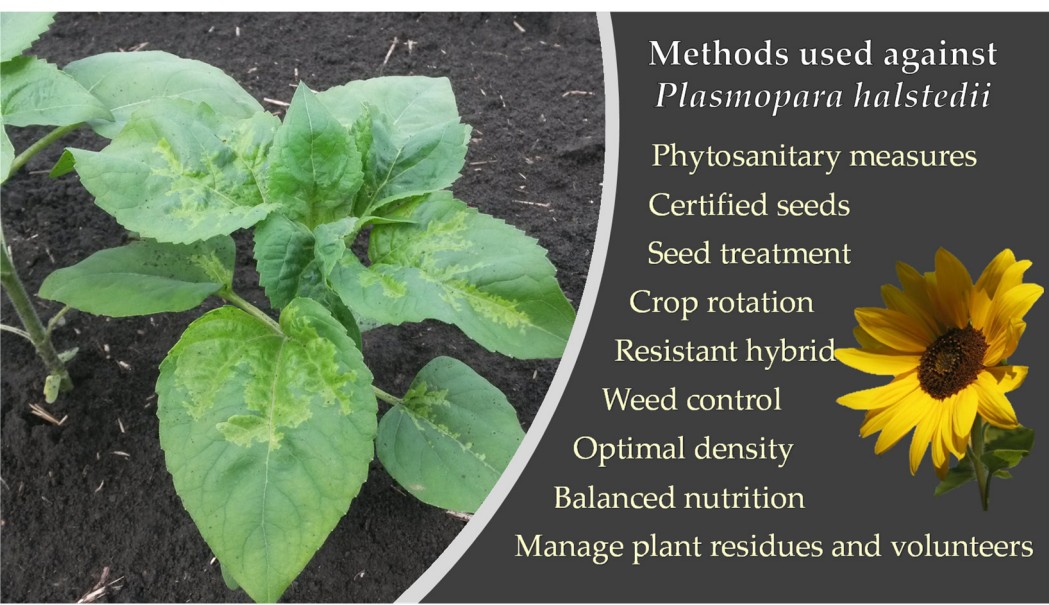

**Figure 2.** Methods used in the integrated disease management of sunflower downy mildew.

*3.2. Phytosanitary Measures on Sunflower Downy Mildew and Seed Control*

The pathogen is seed-borne, so via seed transport, there is a high risk of different pathotypes appearing in sunflower cultivation areas where the hybrids are not resistant to the new variant. This phenomenon is the main reason why *P. halstedii* is a former quarantine, now a regulated non-quarantine pest (RNQP). Phytosanitary measures are, therefore, necessary for preventing the disease.

During the seed production of oil sunflowers, field inspection by plant health officials should ensure that the presence of downy mildew does not exceed the threshold (0%) either for pre-basic or basic and certified seeds [15]. Fields used for seed production have been subject to at least two field inspections during the growing season. All diseased plants have to be removed and destroyed immediately after inspection. A certificate is required to market oil sunflower seeds within the European Community. Additionally, seeds must be treated with an effective plant protection product against all known pathotypes of *P. halstedii.*

Seed coating to prevent the development of early and seed-borne diseases is a crucial component during the seed production of sunflowers. The primary purpose of sunflower seed coating (with a fungicide) is to control downy mildew, as the most relevant early crop

disease [19]. In most cases, certified seeds are already treated with a fungicide. Moreover, at present, farmers have multiple choices to select seeds with different coating techniques (film coating using a polymer, pelleting, or encrusting) that are the most appropriate for their production conditions [38]. Seed pellets and polymers are designed to bind seed-applied treatments, such as pesticides. In addition, they also have an indirect plant protection effect, as they allow uniform and bigger sunflower seed sizes, which helps in seeding [38,39].

There are few active substances registered against *P. halstedii* for seed treatment to date. One of the most efficient compounds for sunflower downy mildew control is mefenoxam (metalaxyl-M), which has long been used for seed coating worldwide [40]. However, from the 1990s onwards, fungicide resistance problems to the compound have been increasingly observed [17,19]. For this reason, the use of mefenoxam in fields has recently been banned [41]. Fortunately, by then, solutions were on the horizon that at present offer similar efficacy to mefenoxam and newer technological innovations. These include, for example, the compounds oxathiapiprolin and benzothiadiazole [32,42].

### 3.3. Combination of Cultural Measures: The Driving Force in IPM against Plasmopara halstedii

Once infected, sunflower plants susceptible to downy mildew cannot be cured, so in addition to phytosanitary measures, other methods for prevention and suppression of disease are much needed. Crop rotation with non-host cultivars is a crucial practice that breaks the pathogen's life cycle. Maintaining a minimum four-year crop rotation is essential because the oospores remain viable for many years in the soil [7]. However, the viability of these resting organs significantly decreases after four years [22].

Crop resistance is the most widely used control method for managing *P. halstedii*. Hybrids resistant to downy mildew have been used successfully for over 50 years [18]. Even now, breeders predominantly use dominant genes against *P. halstedii*; more than 40 dominant resistance genes have been identified [43]. The advantages and disadvantages of vertical resistance conferred by dominant resistance genes are well-known and outlined in several works (see Agrios [44] for an outlook). Since only a few resistant dominant genes have been incorporated into sunflower hybrids that are widely cultivated, the emergence of different *P. halstedii* pathotypes has markedly increased [18]. Breeding programs and genetic research efforts to maximize the diversity of genes in widely cultivated varieties are vital to reducing selection pressure on this pathogen [4,45].

It is noteworthy that hybrids with dominant resistance genes, i.e., optimally 100% protected, can be infected with the pathogen without becoming diseased [46]. Although the pathogen's spread is severely limited, oospores evolve in such hybrids, and persistence is assured. Such a restricted life cycle, especially under a timely narrow crop rotation, leads to the mutation of the pathogen and, thus, to the development of different variants (e.g., pathotypes or fungicide-resistant strains). Furthermore, as hybrids cultivated in the subsequent growing seasons may not be protected against the new strain, resistance breakdown is likely accelerated by timely narrow crop rotation. Therefore, the combination of crop rotation with the use of resistant hybrids is an essential control method against sunflower downy mildew.

In addition to crop rotation, other cultural measures can significantly increase the sustainability of the genetic pool of sunflower hybrids (the lifetime of dominant resistance genes). By creating optimum cultivation conditions, the natural resistance of plants, in addition to genetic resistance to several pests, can be further increased. According to regional recommendations, seeding time and plant density are likely to contribute to the health status of a sunflower stand. Another option in IPM against sunflower downy mildew is to adjust the sowing date according to weather conditions considering local weather factors [7]. Furthermore, balanced nutrition, timely and appropriate weed control, adequate tillage system, and managing crop residues and volunteers are essential to combat *P. halstedii*. Cultural practices that encourage beneficial organisms in the soil are also advisable.

To monitor the pathotype composition of sunflower downy mildew as a preventive method, the regular collection of leaf samples infected with *P. halstedii* is crucial [9]. First, it provides vital information for breeding programs in considering genetic resources. Secondly, farmers' choice of sunflower hybrids is also greatly facilitated by the information on pathotype composition. However, IPM training should be provided to encourage farmers to make knowledgeable choices of sunflower hybrids.

### 3.4. Other Measures to Control Sunflower Downy Mildew

Unlike previous techniques, other IPM solutions are not currently effective or efficient against sunflower downy mildew. Since the spread of the pathogen within the stand is insignificant, control methods applied at the onset of primary symptoms are ineffective [8,9]. Therefore, the management of secondary symptoms is generally not cost-effective. However, chemical control may be necessary to prevent a higher spread of secondary infection, but this can be combined with measures applied to other relevant sunflower diseases (e.g., white rot).

Biological control is not currently a practice in the management of sunflower downy mildew. At the same time, widely used biostimulants with multiple effects are likely to contribute to the control of soil-borne diseases, such as sunflower downy mildew, either indirectly or directly [47].

## 4. Evaluating Current State, Future Challenges, and Perspectives: A SWOT Analysis

The current situation with the challenges and perspectives of integrated pest management against sunflower downy mildew is outlined by a SWOT analysis (Table 1). We identified internal factors, such as strengths and weaknesses, and external factors, such as opportunities and threats, connected to the topic.

**Table 1.** SWOT analysis of the IPM of sunflower downy mildew.

| SWOT ANALYSIS |
|---|
| **INTERNAL FACTORS** |
| **STRENGTHS (+)** |
| ✿ Sunflower is an essential arable crop, so managing its diseases is the subject of intensive research. |
| ✿ Strict regulations during seed production and seed transport limit pathogen spread. |
| ✿ Seed treatment is an environmentally friendly, low-cost method. |
| ✿ Effective seed treatment technologies are available. |
| ✿ Crop rotation as a significant practice in IPM of *P. halstedii*. |
| ✿ Innovative weed management technologies are available. |
| ✿ Major gene (or vertical) resistance conferred by dominant genes is a vital point of the integrated management against *P. halstedii*. |
| **WEAKNESSES (-)** |
| ✿ Latent infection enables the spread of the pathogen. |
| ✿ The results of molecular genetic research are often not quickly translated into practice. |
| ✿ Fungicide resistance against the widely used active ingredients. |
| ✿ Farmers' awareness of new techniques. |
| ✿ Timely narrow crop rotation is used because of profitability. |
| ✿ Only a few resistant dominant genes have been incorporated into sunflower hybrids that are widely cultivated. |
| ✿ Resistance conferred by major genes is quite fragile. |
| **EXTERNAL FACTORS** |
| **OPPORTUNITIES (+)** |
| ✿ New techniques are available to detect the pathogen in the seed (e.g., PCR). |
| ✿ Vast potential in using innovative seed treatment technologies. |
| ✿ Higher diversification of the downy mildew resistance gene pool. |
| ✿ Wild *Helianthus* species serve as a valuable source of resistance. |
| ✿ Introducing small-effect multiple genes into sunflower hybrids. |
| ✿ Vast potential for using alternative crop protection methods against sunflower downy mildew. |
| ✿ Wider advisory and training services for farmers on new research findings. |

**Table 1.** *Cont.*

| SWOT ANALYSIS |
|---|
| **THREATS (-)** |

☼ Climate change and changing political circumstances could negatively impact plant health.
☼ The application of molecular techniques is expensive.
☼ Further spread of the aggressive pathotypes of *P. halstedii* due to inadequate technology transfer.
☼ The pathogen can easily break down the effect of newly developed major genes.
☼ Alternative control methods have only a partial effect if they are not integrated into IPM.

Sunflower is an essential arable crop, so managing its diseases is the subject of intensive research worldwide (*strength*). However, climate change and changing political conditions could directly/indirectly negatively affect the plant health situation of sunflowers (*threat*). Phytosanitary measures are the first line of defense against sunflower downy mildew. In fact, due to strict controls applied by several countries, the spread of the pathogen is mainly limited (*strength*) but not completely inhibited. Furthermore, *P. halstedii* is capable of latent infection, so these plants cannot be screened out during inspections (*weakness*). The latent spread of the pathogen can be traced using molecular techniques, such as polymerase chain reaction (PCR) [48,49], but this has yet to be routinely used (*opportunity*), which is relatively expensive (*threat*). Moreover, because the results of molecular genetic research cannot be rapidly translated into practice (*weakness*), this could further increase the prevalence of *P. halstedii* pathotypes (*threat*).

Plant health regulations [15] require sunflower seeds to be treated with an active ingredient effective against all *P. halstedii* pathotypes. As an essential part of disease management, seed treatment in different ways provides farmers with an environmentally friendly, low-cost solution for protection against sunflower downy mildew (*strength*). To a lesser extent, however, resistance can be developed even against the active ingredients that are widely used for seed treatment (*weakness*), an example being the reduced sensitivity of *P. halstedii* to mefenoxam. There is vast potential in using innovative seed treatment technologies (*opportunity*), such as seed priming [50–52]. However, farmers' awareness of these techniques (*weakness*) should be increased, for example, with policy intervention.

Appropriately integrating sunflowers into the cropping system is reasonably complex regarding their plant sanitary status. Since sunflower has some polyphage pathogens and pathogens that persist in the soil, crop rotation is (or would be) a paramount practice in combatting diseases such as downy mildew (*strength*). However, in many places, mainly for profitability, a timely narrow crop rotation is used (*weakness*), which creates favorable conditions for *P. halstedii*, especially in wet weather [9]. Given this, innovative production systems, such as intercropping and double cropping, must be used to cope with many challenges [53,54].

Plowing is a crucial part of the tillage system in sunflowers because it is essential for the proper development of the sunflower root system [53]. In addition, it helps to manage any infested crop residues. Nevertheless, the widespread use of innovative weed management technologies (Clearfield, Express) in different cropping systems dramatically reduces the chances of *P. halstedii* persisting and spreading (*strength*).

Major gene (or vertical) resistance conferred by dominant genes is one of the most vital points of integrated disease management against *P. halstedii* (*strength*). Unfortunately, only a few resistant dominant genes have been incorporated into sunflower hybrids that are widely cultivated (*weakness*). Moreover, a pathogen can easily break down the effect of dominant genes (*weakness* and *threat*), as it has happened several times in crop production history. Thus, a higher diversification of the downy mildew resistance gene pool is crucial (*opportunity*) [45]. In addition, wild *Helianthus* species serve as a valuable source of resistance genes (*opportunity*) [55].

Horizontal resistance, inherited by small-effect genes rather than dominant genes, has yet to be established in modern hybrids. Indeed, genetic research results are promis-

ing: introducing small-effect multiple genes into sunflower hybrids, reducing selection pressure on the pathogen while retaining other valuable agronomic traits, may be feasible (*opportunity*) [56,57]. Moreover, recent molecular genetic techniques, such as RNA interference, have also become available in sunflowers, although to date only against the Tobacco streak virus [58].

There is massive potential for alternative crop protection methods against sunflower downy mildew (*opportunity*). Biological control has already proven its worth in many areas, and the use of different biopreparations has increased significantly in recent years [37,59]. At present, *Trichoderma* spp. are the most widely used microorganisms in plant disease control in the world [60]. For example, Nagaraju et al. [61] identified that *Trichoderma harzianum* has contributed significantly to reduced disease incidence caused by *P. halstedii* in a susceptible sunflower in both greenhouse and field experiments. In addition, microbial-based biostimulants are widely available for farmers to improve the quality and quantity of their crops [47]. However, at present, the most effective solutions against *P. halstedii* are those that target the treatment of sunflower seeds.

Bio-priming of sunflower seeds, i.e., seed priming combined with different resistance inducers, may help to protect against downy mildew, as it proved effective against Alternaria blight in sunflowers [50]. Indeed, in earlier studies, resistance inducers were efficient against sunflower downy mildew [62,63]. The application of entomopathogenic fungi [11,64], mycorrhizal fungi [65,66], and botanical pesticides, such as NeemAzal [67], is also promising against *P. halstedii* and sunflower diseases. Moreover, the recent discovery of the *Plasmopara halstedii virus*, which causes hypovirulence in the sunflower downy mildew pathogen, could open up new perspectives in plant protection in the future [68]. Finally, alternative control methods have only a partial effect if they are not integrated into IPM (*threat*). Broader advisory and training services for farmers on new research findings and their practical implementation are also necessary (*opportunity*).

## 5. Conclusions

At present, sunflower production is facing many challenges. Climate change, the current economic and political situation, and the increasing need to protect the environment and decrease the risk of synthetic pesticides present many issues to be addressed. An integrated pest management approach against major sunflower diseases, such as sunflower downy mildew, is essential. In addition to traditional tools (e.g., genetic resistance, seed treatment, crop rotation, and balanced nutrition), the implementation of new scientific advances (e.g., seed priming, RNA interference, horizontal resistance, and biostimulants) will enable sustainable sunflower production in the future. In the meantime, however, careful consideration of the different protection methods and continuous evaluation of the system is fundamental.

**Author Contributions:** Conceptualization, R.B., J.K., Z.P. and K.K.; methodology, R.B. and J.K.; software, R.B.; validation, R.B., J.K., Z.P. and K.K.; formal analysis, R.B., J.K. and K.K.; investigation, R.B., J.K., Z.P. and K.K.; data curation, R.B., J.K. and Z.P.; writing—original draft preparation, R.B. and J.K.; writing—review and editing, R.B., J.K., Z.P. and K.K.; visualization, R.B., Z.P. and K.K.; supervision, R.B. and J.K.; project administration, R.B. and J.K.; funding acquisition, R.B. and J.K. All authors have read and agreed to the published version of the manuscript.

**Funding:** This research was funded by the Biological Sciences PhD School and the Plant Sciences PhD School of the Hungarian University of Agriculture and Life Sciences.

**Data Availability Statement:** Not applicable.

**Acknowledgments:** The authors are grateful to Mihály Perczel (PlasmoProtect) and Ferenc Virányi for their dedicated work and significant support.

**Conflicts of Interest:** The authors declare no conflict of interest. The funders had no role in the design of the study; in the collection, analyses, or interpretation of data; in the writing of the manuscript; or in the decision to publish the results.

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
