# Peer review of "Placing Management of Sunflower Downy Mildew (Plasmopara halstedii (Farl.) Berl. et de Toni) under an Integrated Pest Management (IPM) System Approach: Challenges and New Perspectives"

_agronomy, doi:10.3390/agronomy13041029_

Round 1
Reviewer 1 Report
Rita Bán et al., provide the opinions about management of sunflower downy mildew. The manuscript is not well-written and thus should be improved.
1. Provide full name of ‘IPM’ in the title.
2. What is SWOT in the abstract?
3. It is not appropriate to mention the conflicts between Ukraine and Russia.
4. L38, keep the same name of the pathogen throughout the manuscript.
5. L57, what is the variability?
6. L58, there should be citations.
7. It is better to provide scale bars in Figures.
Reviewer 2 Report
Manuscript name: Placing management of sunflower downy mildew (Plasmopara halstedii (Farl.) Berl. et de Toni) under an IPM system approach: challenges and new perspectives
This article is significant because it uses SWOT analysis to evaluate the integrated pest management system against downy mildew in sunflower. However, the following changes are required.
1. The article's major problem is that the reference numbers are mixed up. This is because reference 17 is written in the explanation of reference 16 on line 380. On line 117, for example, reference 33 belongs to reference 34 in the reference list. Again, reference 34 on line 121 corresponds to reference 35 in the reference list. Please review the reference numbers following 16 throughout the article.
2. Line 86: Please use the term "on abaxial surface of leaves" instead of "on the back"
3. Line 306-310: Trichoderma spp. has been studied in the control of sunflower downy mildew. This paragraph should also include these studies.
Reviewer 3 Report
Whole article is written as a review of known information supplemented by a SWOT analysis on IPM of sunflower downy mildew. Corresponding author is a reputable specialist in this filed, parts are logically divided and incorporating a big amount of information concerning the topic and English language is correct. Only one topic I would recommend to slightly mention in the review, namely problematic of Plasmopara halstedii virus (e.g. Grasse et al., 2013, https://doi.org/10.1016/j.fgb.2013.05.009).
I recommend the acceptance of the manuscript.
Reviewer 4 Report
The manuscript was well-prepared. The opinions on the IPM against sunflower downy mildew are constructive. I have no questions.
Reviewer 5 Report
none
Round 2
Reviewer 1 Report
This manuscript can be accepted.
Reviewer 2 Report
Manuscript name: Placing management of sunflower downy mildew (Plasmopara halstedii (Farl.) Berl. et de Toni) under an IPM system approach: challenges and new percpectives.
The article is significant because it uses SWOT analysis to evaluate the intengrated pest management system against downy mildew in sunflower.
The authors have followed all my suggestions. I thank them for this.
